# Rapid On-Site Evaluation Performed by an Interventional Pulmonologist: A Single-Center Experience

**DOI:** 10.3390/jpm14070764

**Published:** 2024-07-18

**Authors:** Emanuela Barisione, Carlo Genova, Matteo Ferrando, Maurizio Boggio, Michele Paudice, Elena Tagliabue

**Affiliations:** 1Interventional Pulmonology Unit, IRCCS Ospedale Policlinico San Martino, 16132 Genoa, Italy; emanuela.barisione@hsanmartino.it (E.B.); elena.tagliabue@hsanmartino.it (E.T.); 2Academic Oncology Unit, IRCCS Ospedale Policlinico San Martino, 16132 Genoa, Italy; 3Department of Internal Medicine and Medical Specialties (DIMI), University of Genoa, 16132 Genoa, Italy; 4Riabilitative Pulmonology Unit, Ospedale di Sestri Levante, 16039 Sestri Levante, Italy; matteo.ferrando@asl4.liguria.it; 5Anatomic Pathology Unit, IRCCS Ospedale Policlinico San Martino, 16132 Genoa, Italy; maurizio.boggio@hsanmartino.it; 6Department of Surgical Sciences and Integrated Diagnostics (DISC), University of Genoa, 16132 Genoa, Italy; michele.paudice@unige.it; 7University Pathology Unit, IRCCS Ospedale Policlinico San Martino, 16132 Genoa, Italy

**Keywords:** ROSE, bronchoscopy, EBUS, cytopathologist, pulmonologist, diagnosis, lung cancer, lymph nodes

## Abstract

Background: Rapid On-Site Evaluation (ROSE) during bronchoscopy allows us to assess sample adequacy for diagnosis and molecular analyses in the context of precision oncology. While extemporaneous smears are typically evaluated by pathologists, their presence during bronchoscopy is not always possible. Our aim is to assess the concordance between ROSE performed by interventional pulmonologists and cytopathologists. Methods: We performed ROSE on 133 samples collected from 108 patients who underwent bronchoscopy for the diagnosis of suspect thoracic findings or for mediastinal lymph node staging (May 2023–October 2023). Randomly selected smears (one for each collection site) were independently evaluated for adequacy by a pulmonologist and a pathologist to assess the concordance of their evaluation. Results: Among 133 selected smears evaluated by a pulmonologist and pathologist, 100 were adequate for both, 10 were inadequate for both and 23 were discordant; hence, global concordance was 82.7%; Cohen’s Kappa was 0.385, defining fair agreement. Concordance was similar irrespective of sample collection site (lymph nodes vs. pulmonary lesions; *p* = 0.999) and among samples which were considered adequate or inadequate by the pulmonologist (*p* = 0.608). Conclusions: Trained pulmonologists can evaluate the appropriateness of sampling with good concordance with cytopathologists. Our work supports autonomous ROSE by pulmonologists where pathologists are not immediately available.

## 1. Introduction

The utility of Rapid On-Site Evaluation (ROSE) during endoscopic procedures has been a controversial topic for several years, but there is increasing evidence in support of its usefulness [1]. With ROSE, the cytological sample obtained during the bronchoscopic procedure can be instantaneously analyzed, thus allowing for immediate quality assessment. Indeed, the smear is analyzed on time, and once its appropriateness is confirmed or denied, the interventional pulmonologist can be certain of its biopsy site or switch it. This allows us, in addition to saving time and money, to avoid repeating the procedure.

The main role of the pathologist is to provide a histologic definition of the tumor. Cytologic preparations, using hematoxylin–eosin or specific staining, combined with immunohistochemical (IHC) staining allow us to define the tumor’s histology. Cell blocks are of great utility and should always be set up from cytologic material during bronchoscopic procedures. Today, innovative sequencing methods are available, such as next-generation sequencing (NGS), which allows for the simultaneous study of multiple genes from a single tumor sample, provided that such tissue is adequately representative of the neoplasm [2]. The increasing availability of multi-genic molecular methods has allowed for a more extensive profiling of neoplasms, with critical therapeutic implications. However, a possible limitation to the extensive characterization of neoplasms is the adequacy of analyzable tumor tissue, especially in the case of neoplasms that are difficult to biopsy, such as lung cancer. Endobronchial ultrasound transbronchial needle aspiration (EBUS-TBNA) is a key procedure in interventional pulmonology, as it provides smears and cell blocks (pellets of cells incorporated in paraffin and treated like surgical specimens), which are used by the pathologist and molecular biologist for diagnosis and subsequent molecular analyses [3]. Technically, the presence of a dedicated pathologist in the endoscopy room allows for increased diagnostic yield by assessing the adequacy of the specimens during the procedure [4]. However, due to increasing work schedules and procedures, and due to increasing personnel limitations, it may not be possible to ensure the presence of a pathologist during each bronchoscopy, with a potential negative impact on the diagnostic yield of endoscopic procedures. Hence, our hypothesis is that an interventional pulmonologist with adequate training may be able to autonomously perform ROSE during endoscopic procedures without external aid, with good results in terms of adequacy assessments.

## 2. The Aim of This Study

The aim of our study is to compare the performance of ROSE autonomously performed by a trained interventional pulmonologist and by a cytopathologist, both with experience in ROSE, in a series of endoscopic diagnostic/staging procedures performed within the interventional pulmonology unit of a Comprehensive Cancer Center.

## 3. Methods

For this study, we included a cohort of consecutive patients who were deemed candidates for endoscopic procedures to achieve diagnosis and/or mediastinal staging for suspected pulmonary neoplasm in the timespan from 1 May 2023 to 31 October 2023, as part of a wider, currently ongoing monocentric study involving multimodal assessments for patients with thoracic malignancies, registered by the Regione Liguria Local Ethics Committee (P.R. 191REG2015 (v4.0 19 July 2021)). Eligible patients were selected on the basis of radiological and/or metabolic imaging (computerized tomography and positron emission tomography, respectively), in which pulmonary lesions or mediastinal lymph nodes were considered accessible for endoscopic assessment.

The endoscopic procedures were performed by an interventional pulmonologist with acknowledged experience in bronchoscopy and EBUS, who underwent specific training at another institution under the mentoring of expert cytopathologists, which included a final exam for acquiring ROSE certification. Subsequently, the same pulmonologist had additional informal training for three months with an expert cytopathologist from our institution, involving both theory and actual cases.

The material collected through the endoscopic procedure was directly smeared over glass slides and colored by rapid stains. In our practice, we adopted Diff-Quick stain on unfixed air-dried smears, which allowed for the evaluation of sampling adequacy on light microscopy in a few minutes. We bathed the labeled slides 5 times, each for 1–2 s, firstly in the methanol-based fixative solution (blue), then in the eosin-based acid solution (red) and finally in the thiazine and methylene blue-based basic solution (blue). After washing with distilled water at pH 7, the sample was left to air-dry to facilitate the elimination of excess water. 

In order to be considered adequate, a ROSE sample needed to fulfill specific criteria, as follows. When performed on a lymph node site, a sample was considered adequate in the presence of at least one of the following features: (a) a lymphocyte count of 20–60 for field; (b) 90% presence of anthracotic particulate; and (c) presence of neoplastic cells. The criteria employed for establishing ROSE adequacy on lymph nodal lesions were based on published literature data. Choi et al. reported an algorithm according to which the first adequacy criterion was based on a core size ≥ 2 cm on gross inspection; in the case of lesser samples, the aforementioned criteria are generally taken into account. For our study, we decided to consider the three criteria based on microscopic assessment, rather than defining the sample adequate on the basis of core size, in order to challenge the actual ability of the pulmonologist to assess the adequacy of lymph nodal samples at microscopy observation [5]. In the case of ROSE performed on a non-lymph nodal site, adequacy was assessed through the presence of neoplastic cells and/or ectopic tissue (not expected at that site). Notably, while the criteria for the adequacy of lymph nodal samples are well defined, as stated above, the adequacy criteria for non-lymph nodal sites reported in the literature are significantly less specific, as adequate samples are generally reported as “relatively abundant and well represented lesion material” [6,7]; hence, we based our definition of the adequacy of non-lymph nodal lesions on the observation of abnormal material, either neoplastic or non-neoplastic. 

The interventional pulmonologist autonomously evaluated the adequacy of multiple slides for each procedure, and this determination was used to decide whether to proceed with sample collection for final analysis or to switch sampling site. During ROSE, one of the slides for each procedure was randomly selected (either adequate or inadequate) and subsequently submitted to a pathologist with experience in cytopathology and ROSE, blinded to the pulmonologist’s decision over each individual slide. The concordance in terms of the adequacy vs. inadequacy of the selected slides between the pulmonologist and pathologist was assessed by calculating Cohen’s Kappa. More specifically, Cohen’s Kappa values were categorized as follows: 0.01–0.20 slight agreement; 0.21–0.40 fair agreement; 0.41–0.60 moderate agreement; 0.61–0.80 substantial agreement; and 0.81–1.00 almost perfect or perfect agreement.

## 4. Results

Globally, 108 consecutive patients underwent endoscopic diagnostic/staging procedures, for a total of 133 site collections. The mean age of the patients was 68 years (range: 35–89), and the male/female ratio was 67/41; smoking status was the following: 25 never smokers, 48 former smokers and 35 active smokers. We performed 1–4 passages in TBNA with EBUS for each patient in the target lesions (average: 2 passages), for an average duration of the procedure of 24 min (range: 22–36 min), including ROSE reading. Collection sites are reported in Table 1.

For 74 out of 108 patients (68.5%), the endoscopic procedure was positive (in other terms, a diagnosis of malignant or benign acknowledged disease was achieved). Notably, among the 34 patients with negative endoscopy, most (*n* = 26) underwent EBUS for mediastinal lymph node assessment; hence, a negative result for either benign or malignant disease was not considered a false negative. Finally, eight patients (7.4%) had a negative sample after diagnostic approach to a pulmonary lesion. Specific diagnostic findings are reported in Table 2. Notably, neither the number of samples nor the total number of needle passages was associated with differences in terms of final biopsy performance (Mann–Whitney *p* = 0.8736 and *p* = 0.3972, respectively). 

When we assessed the concordance of ROSE between the interventional pulmonologist and cytopathologist on one randomly selected slide for each collected sample within the full set of 133 procedures, the positive ROSE calls according to the pulmonologist were 121/133 (90.9%), while the positive ROSE calls according to the pathologist were 102/133 (76.7%). More specifically, we observed the following results: (a)A total of 100 slides were considered adequate by both the pulmonologist and the pathologist.(b)A total of 10 slides were considered not adequate by both the pulmonologist and the pathologist.(c)A total of 2 slides were considered not adequate by the pulmonologist and adequate by the pathologist.(d)A total of 21 slides were considered adequate by the pulmonologist and not adequate by the pathologist.

Hence, with regard to the full set of 133 samples, agreement was 82.7%, and Cohen’s Kappa was equal to 0.385 (Confidence Interval = 0.207 to 0.589; Standard Error = 0.098), resulting in fair agreement.

When we assessed the concordance of ROSE between the interventional pulmonologist and cytopathologist within the sub-set of samples involving lymph nodes (*n* = 101), we observed the following results:(a)A total of 79 slides were considered adequate by both the pulmonologist and the pathologist.(b)A total of 5 slides were considered not adequate by both the pulmonologist and the pathologist.(c)A total of 2 slides were considered not adequate by the pulmonologist and adequate by the pathologist.(d)A total of 15 slides were considered adequate by the pulmonologist and not adequate by the pathologist.

Hence, with regard to the sub-set of 101 lymph nodal samples, agreement was 83.2%, and Cohen’s Kappa was equal to 0.298 (Confidence Interval = 0.066 to 0.531; Standard Error = 0.119), resulting in fair agreement.

Finally, when we assessed the concordance of ROSE between the interventional pulmonologist and cytopathologist within the sub-set of samples involving lung lesions (*n* = 32), we observed the following results:(a)A total of 21 slides were considered adequate by both the pulmonologist and the pathologist.(b)A total of 5 slides were considered not adequate by both the pulmonologist and the pathologist.(c)A total of 6 slides were considered not adequate by the pulmonologist and adequate by the pathologist.(d)A total of 0 slides were considered adequate by the pulmonologist and not adequate by the pathologist.

Hence, with regard to the sub-set of 32 pulmonary lesion samples, agreement was 81.3%, and Cohen’s Kappa was equal to 0.522 (Confidence Interval = 0.220 to 0.825; Standard Error = 0.154), resulting in moderate agreement. Hence, we identified no differences in terms of concordance between lymph nodes and pulmonary lesions (Fisher *p* value = 0.608), as reported in Figure 1.

Subsequently, we explored the possible differences in concordance among samples based on adequacy, as defined by the interventional pulmonologist, to define whether the concordance with the pathologist varied among samples which were considered adequate or inadequate by the pulmonologist; notably, no significant difference was observed (Fisher *p* value = 0.999; Figure 2). 

Finally, we did not perform a correlation analysis of each ROSE sample with the final diagnosis, as inadequate samples (by pulmonologist judgment) immediately led to additional sampling in different sites.

## 5. Discussion

The literature of the past five years is quite extensive regarding the use of ROSE in the endoscopic room, generally supporting its role as a baseline integration of endoscopic procedures, consistent with our experience [8,9,10,11,12,13,14,15,16,17,18,19,20,21,22,23,24,25,26,27]. ROSE also appears very useful during the cryobiopsy of peripheral lung lesions due to its high specificity and positive predictive value [28]. Furthermore, ROSE has also been used in granulomatous diseases [29]. In the literature, it has also been pointed out that the amount of sample employed for ROSE is exiguous; therefore, virtually no material is subtracted for subsequent assessments, such as molecular analyses for the identification of actionable genic alterations [30]. Additionally, ROSE has been reported to also be useful in procedures involving electromagnetic navigation bronchoscopy (ENB), with a high positive impact on the diagnosis. For instance, in the prospective, multicenter NAVIGATE study, the impact of ROSE was assessed in the context of an extensive multimodal strategy (including biopsy forceps, cytology brush, aspirating needle, triple-needle cytology brush, needle-tipped cytology brush, core biopsy system and bronchoalveolar lavage); notably, a positive ROSE call for a malignant lesion reduced the number of tools employed during each procedure [31]. 

Although the usefulness of ROSE during endoscopic sessions is well-acknowledged, this technique is hardly established in common clinical practice. The main reason for this occurrence is represented by organizational difficulties caused by the number of relevant procedures and a lack of sufficient personnel to ensure the presence of a pathologist during each bronchoscopy [32]. Of note, the national cross-sectional study conducted in China by Haidong Huang et al. pointed out the lack of a cytopathologist figure limiting the use of ROSE to only 28 out of 347 included centers [33]. In order to address this issue, innovative methods have been explored. A recent study by Dou et al. investigated the use of a chip able to distinguish malignant from benign lung nodules with the indentation of methylation sites. This panel combined with cytology, ROSE and histology brings the sensitivity of the panel to 90.8% and 95.8% in bronchial lavage and bronchial brushing samples, respectively, and 100% in lavage and brushing samples. While brushing and bronchial lavage are not currently used as diagnostic baseline techniques in our center, the study by Dou et al. shows that we are currently moving toward a combination of techniques and analyses to increase sensitivity and specificity, with ROSE being one of the main tools [34]. A very practical attempt to overcome the technical difficulty associated with the presence of a pathologist in the endoscopic room was reported by Damaraju et al.; in their study, images of the samples were shared with a cytopathologist via instant messaging, thus allowing for specialist evaluation without needing the physical presence of the pathologist during endoscopy. The reported results are encouraging, but in spite of constant improvement, this practice still requires a pathologist to be available for immediate assessment [35]. 

Notably, there are enough data to suggest that well-trained pulmonologists may be able to perform ROSE, but this practice is still seldom employed, and no wide diffusion across hospitals has been observed so far [36], its main limitation being the need for specific training. While there are acknowledged courses, such as the certified training programs held by the European Respiratory Society [6,37,38], there is a global inconsistency in terms of training across different studies. Indeed, the reported training duration varies, typically ranging from 1 to 6 months [18,36,39,40], and in some cases, the specific duration and modalities of training are not specified [41,42]. In this regard, the authors strongly encourage the development of organized training programs with consistent modalities, durations and examinations for achieving board certification; furthermore, re-training should be deemed necessary, especially when the interventional pulmonologist cannot perform an adequate number of ROSE assessments in a pre-determined amount of time, which has yet to be clearly defined.

One of the endpoints of our study was to determine the concordance rate for adequacy assessment between pulmonologists and pathologists. A concordance value of 82.7% suggests that a trained pulmonologist can evaluate the appropriateness of the sampling with a similar performance of the cytopathologist. Notably, in our study, the concordance between the pulmonologist and pathologist was not influenced by the collection site (lymph nodes vs. pulmonary lesions); similarly, the judgment of adequacy vs. inadequacy by the pulmonologist was not associated with significant differences in terms of concordance with the pathologist. Taken together, all these data suggest global consistency in concordance across the collected specimens. Surprisingly, the pulmonologist was globally more likely to call a positive ROSE compared to the pathologist in this specific set of 133 slides (90.9% vs. 76.7%). This difference may be partially associated with the study procedure, since the pulmonologist’s ROSE was performed during the endoscopic procedure, with direct and comprehensive knowledge of the approached collection sites, whereas the pathologist’s ROSE was performed retrospectively, only on randomly selected slides provided by the pulmonologist. While this design may appear as a limitation, it was deemed necessary in order to properly perform the bronchoscopy/EBUS without excessively increasing the procedure time, which would be poorly tolerated by the patients. Furthermore, we observed a good global performance in terms of diagnostic yield when ROSE was dependent from the pulmonologist’s judgment, with most biopsies on pulmonary lesions having diagnostic results, whereas negative results for benign/malignant disease were mostly found among lymph nodal staging procedures, as expected. 

Notably, since the adequacy criteria for lymph node samples are more precise compared to criteria for pulmonary lesions, we were expecting differences in terms of concordance according to the sampling site, as the pulmonologist had fewer reference points for defining an adequate pulmonary lesion, and the experience of a cytopathologist might have an edge; in fact, no significant difference was observed in terms of concordance between lymph nodal and non-lymph nodal lesions, suggesting a similar performance of the pulmonologist irrespective of collection site. One possible explanation can be found in the concept that non-lymph nodal lesions simply require less complex criteria since the adequacy of sample is defined by the presence of abnormal cells, either neoplastic or non-neoplastic, while lymph nodes actually need more defined criteria due to their intrinsic complexity. Additionally, with regard to discordant cases between the pulmonologist and pathologist, since we were expecting the pulmonologist to be less likely to call adequate samples compared to the pathologist, due to lower experience, we have developed some speculations on the increased cases of adequate lymph nodal ROSE calls made by the pulmonologist: (1) the pulmonologist might have been less confident in discriminating between lymphocytes actually belonging to the assessed lymph node (“true positive” for adequacy) and circulating lymphocytes collected through fine-needle aspiration where bleeding occurred (“false positive” for adequacy); (2) similarly, the presence of necrosis in the collected sample of either a lymph node or a pulmonary lesion might have been mistakenly interpreted as the presence of lymphocytes or neoplastic cells, thus leading to a positive ROSE call. 

Previous studies have already reported the accuracy of ROSE performed by trained interventional pulmonologists [6], but the evidence on the comparison of pulmonologists and pathologists is more limited. In this regard, the concordance of ROSE between pulmonologists and pathologists has already been evaluated in a cohort of 84 patients who underwent endoscopy for mediastinal lymph nodal assessment, achieving an overall substantial agreement equal to 81% [40]. Compared to this experience, our analysis included a larger patient population and, most importantly, included a relevant sub-set of patients with pulmonary lesions, for which the agreement between the pulmonologist and pathologist was 81.3%, similar to lymph nodes. In another experience, ROSE concurrently performed by a pulmonologist and pathologist in a population of 104 patients, for a total of 164 aspirated sites (including endoscopic as well as percutaneous aspirations), achieved an agreement equal to 86% but required the concurrent presence of the pulmonologist and the pathologist during each procedure [37]. In this context, the value of our study lies in the relevant number of patients and lesions, in the presence of mediastinal lymph nodes as well as pulmonary lesions; additionally, in the study plan, the procedure was carried out on the basis of the pulmonologist’s judgment. While on one hand, this approach may have led to the reduced tendency of the pathologist to call positive ROSE slides, as the call was performed retrospectively, the results in terms of the final diagnostic yielding further support the use of ROSE performed by trained pulmonologists in daily oncology practice. We also acknowledge that a possible limitation of our study can be found in the choice to randomly select one slide from each collection site for concordance assessment, rather than providing all the slides to the pathologist; however, the random selection was performed in order to focus the pathologist’s attention on each single slide and to prevent possible influences depending on a higher number of slides for difficult collection sites (hence more likely to have at least some inadequate samples). Additionally, we did not compare ROSE performed by the pulmonologist with the final diagnostic results, as our aim was to focus on the comparison between the pulmonologist and pathologist on individual sample slides. In our opinion, the assessment of the performance and future possibilities of ROSE deserves further studies, and in this regard, a prospective, multicentric study assessing the performance of ROSE calls performed by pulmonologists of different institutions may help in understanding the potential advantages and pitfalls of the systematic practice of ROSE calls performed by trained pulmonologists, eventually with the additional aim of addressing the possible impact of different training modalities among pulmonologists.

In conclusion, our work supports the use of ROSE performed by a trained pulmonologist during bronchoscopy/EBUS, with acceptable agreement with the pathologist’s assessment and good outcomes in terms of diagnostic yield. In the authors’ opinion, interventional pulmonology units should support the training of personnel in order to ensure that ROSE can be autonomously performed by pulmonologists in daily clinical practice, resulting in the optimized diagnostic yield of bronchoscopy/EBUS when cytopathologists are not available or eventually allowing these specialists to commit to different, more specific tasks, while ROSE is effectively performed by pulmonologists.

## Figures and Tables

**Figure 1 jpm-14-00764-f001:**
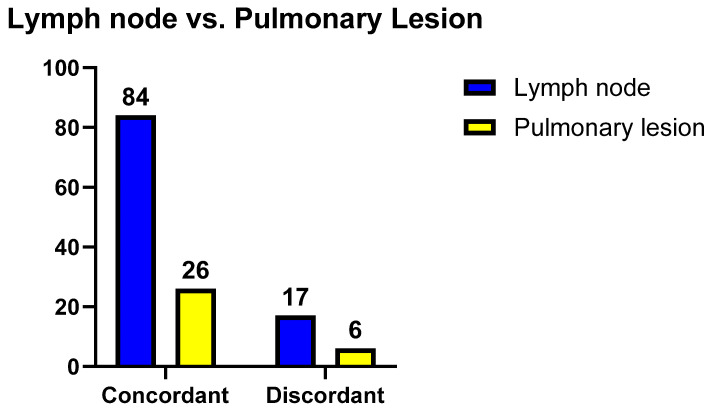
Contingency tables comparing concordance between pulmonologist and pathologist based on sampling site (Fisher *p* value = 0.7926).

**Figure 2 jpm-14-00764-f002:**
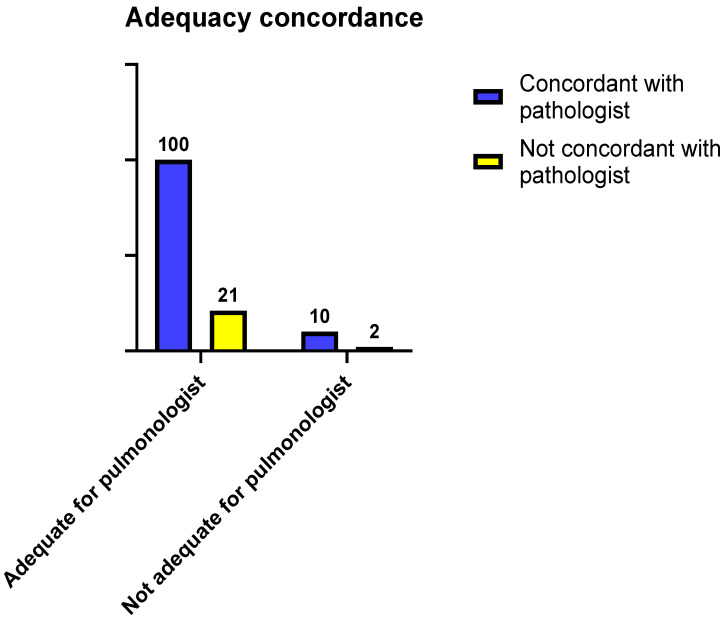
Contingency tables comparing concordance with pathologist based on adequacy as determined by interventional pulmonologist (Fisher *p* value = 0.999).

**Table 1 jpm-14-00764-t001:** The distribution of collection sites among the 133 procedures.

Collection Site		N
Lymph Nodes	All stations	101
	Station 7	40
	Station 4R	32
	Station 11L	6
	Station 4L	5
	Station 11R	7
	Station 2R	4
	Station 10L	3
	Station 10R	4
Pulmonary lesions	All lesions	32
	Right upper lobe	9
	Left upper lobe	8
	Right lower lobe	6
	Left lower lobe	4
	Middle lobe	2
	Lingula	2
	Intermediate right bronchus (suspect recurrence after surgical resection)	1

**Table 2 jpm-14-00764-t002:** The final pathological diagnosis for all the 108 patients who underwent endoscopic procedures.

Diagnostic Finding		N
Primary pulmonary cancer	Pulmonary adenocarcinoma	34
	Squamous-cell lung cancer	9
	Small-cell lung cancer	7
	Not otherwise specified lung cancer	4
	Poorly differentiated non-small-cell lung cancer	3
	Large-cell neuroendocrine pulmonary cancer	1
Other malignancies excluding primary pulmonary cancer	Lymphoma	5
	Metastasis from breast cancer	2
	Metastasis from endometrial cancer	1
	Metastasis from melanoma	1
	Solitary fibrous tumor	1
Non-neoplastic diagnoses	Dysplasia	1
	Granulomatosis	4
	Idiopathic pulmonary fibrosis	1
Negative		34

## Data Availability

The original contributions presented in the study are included in the article, further inquiries can be directed to the corresponding author.

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
