# Peer review of "Rapid On-Site Evaluation Performed by an Interventional Pulmonologist: A Single-Center Experience"

_jpm, 2024, doi:10.3390/jpm14070764_

Round 1

Reviewer 1 Report

Comments and Suggestions for Authors

The manuscript presents a valuable study on the effectiveness of interventional pulmonologists performing Rapid On-Site Evaluation (ROSE) during bronchoscopy compared to cytopathologists. The findings indicate a high concordance rate and suggest that trained pulmonologists can effectively perform ROSE, potentially improving diagnostic workflows.

Please address the following questions:

1.  please provide more detailed information on the training protocol to ensure reproducibility and standardization across different centers for performing ROSE, add more references.

2. Please elaborate on the criteria used to determine the adequacy of samples. Specifically, how were the thresholds for lymph node and pulmonary lesion established, and do these criteria align with existing standards in the literature?

3. How did you address the 23 discordant results between the pulmonologists and pathologists? Were there any identifiable patterns or factors contributing to these discrepancies, and what steps could be taken to minimize them in future studies?

4. The study mentions no significant difference in concordance between lymph node and pulmonary lesion samples. Can you provide a more detailed analysis of the possible reasons behind the variation in agreement rates?

Author Response

Dear Editor and Reviewers,

We wish to thank you for the rapid and comprehensive revision of our manuscript, entitled “RAPID ON-SITE EVALUATION PERFORMED BY THE IN-TERVENTIONAL PULMONOLOGIST: A SINGLE-CENTRE EXPERIENCE”.

We have read your comments and tried to provide answers and revisions to the best of our knowledge and capabilities. In our opinion, with the aid of your comments, the current version of the article has generally improved, making it more useful for any potential reader.

Here you can find the comments and our answers, with reference to the sections of the manuscript which have been modified.

Once again, we wish to thank you for your consideration of our work.

Sincerely,

On behalf of all the Authors,

Dr. Carlo Genova

University of Genova, Italy

REVIEWER 1

  1. please provide more detailed information on the training protocol to ensure reproducibility and standardization across different centers for performing ROSE, add more references.

ANSWER: thank you for your comment. Indeed, information on the training protocol was initially lacking from our manuscript. Hence, we reported the training followed by our interventional pulmonologist in the Methods section, and we also included some observations on the global inconsistency of training across different published studies, with additional literature reference.

More specifically, we modified the following sections:

METHODS (lines 93-98): “The endoscopic procedures were performed by an interventional pulmonologist with acknowledged experience in bronchoscopy and EBUS, who underwent a specific training at another institution under mentoring of expert cytopathologists, which included a final exam for acquiring certification of ROSE performer. Subsequently, the same pulmonologist had an additional, informal training for three months with an ex-pert cytopathologist from our institution, involving both theory and actual cases.”

DISCUSSION (lines 267-276): “While there are acknowledged courses, such as the certified training programs held by the European Respiratory Society (6, 37, 38), there is a global inconsistency in terms of training across different studies. Indeed, the reported training duration varied, typically ranging from 1 to 6 months (18, 39, 40, 41), and in some cases, the specific duration and modalities of training are not specified (42, 43). To this regard, the authors strongly encourage the development of organized training programs with consistent modalities, duration, and examinations for achieving a board certification; further-more, a re-training should be deemed necessary, especially when the interventional pulmonologist cannot perform an adequate number of ROSE assessments in a pre-determined amount of time, which has to be clearly defined yet.”

  1. Please elaborate on the criteria used to determine the adequacy of samples. Specifically, how were the thresholds for lymph node and pulmonary lesion established, and do these criteria align with existing standards in the literature?

ANSWER: Thank you for this observation. We included more information regarding the criteria employed for adequacy, both for lymph nodes and for pulmonary lesions. Altogether with the information, we included relevant references supporting the employed criteria. The information was added in the METHODS section, as it follows.

METHODS (lines 107-125): “In order to be considered adequate, a ROSE sample needed to fulfil specific criteria, as it follows. When performed on a lymph node site, a sample was considered adequate in presence of at least one of the following features: a) a lymphocyte count of 20-60 for field; b) 90% presence of anthracotic particulate; c) presence of neoplastic cells. The criteria employed for establishing ROSE adequacy on lymph nodal lesions were based on published literature data. Choi et al. reported an algorithm according to which the first adequacy criterion was based on core size ≥ 2 cm on gross inspection; in case of lesser samples, the aforementioned criteria are generally taken into account. For our study, we decided to consider the three criteria based on microscopic assessment, rather than defining the sample adequate on the basis of core size, in order to challenge the actual ability of the pulmonologist to assess adequacy of lymph nodal samples at microscopy observation (5). In the case of ROSE performed on a non-lymph nodal site, adequacy was assessed through the presence of neoplastic cells and/or ectopic tissue (not expected at that site). Notably, while the criteria for adequacy of lymph nodal samples are well defined, as stated above, the adequacy criteria for non-lymph nodal sites reported in literature are significantly less specific, as adequate samples are generally reported as “relatively abundant and well represented lesion material” (6, 7); hence, we based our definition of adequacy of non-lymph nodal lesion on the observation of abnormal material, either neoplastic or non-neoplastic.”

  1. How did you address the 23 discordant results between the pulmonologists and pathologists? Were there any identifiable patterns or factors contributing to these discrepancies, and what steps could be taken to minimize them in future studies?

ANSWER: Thank you for the comment. Addressing the differences between pulmonologist and pathologist was no easy task, as initially we were expecting the pulmonologist to be more reluctant in positive ROSE calls compared to the pathologist. However, we made some speculations regarding how some cases were considered adequate by the pulmonologist and not adequate by the pathologist, which made up for most discordant cases, actually. We reported our observations in the discussion section, as it follows:

DISCUSSION (lines 308-318): “Additionally, with regards to discordant cases between pulmonologist and pathologist, since we were expecting the pulmonologist to be less likely to call adequate samples compared to the pathologist, due to lower experience, we have developed some speculations on the increased cases of adequate lymph nodal ROSE calls made by the pulmonologist: 1) the pulmonologist might have been less confident in discriminating between lymphocytes actually belonging to the assessed lymph node (“true positive” for adequacy) and circulating lymphocytes collected through fine needle aspiration where bleeding has occurred (“false positive” for adequacy); 2) similarly, the presence of necrosis in the collected sample of either a lymph node or a pulmonary lesion might have been mistakenly interpreted as presence of lymphocytes or neoplastic cells, thus leading to a positive ROSE call.”

  1. The study mentions no significant difference in concordance between lymph node and pulmonary lesion samples. Can you provide a more detailed analysis of the possible reasons behind the variation in agreement rates?

ANSWER: Thank you for this comment. Indeed, we were looking for differences in terms of concordance between lymph nodes and pulmonary lesions, but such differences were not significant, as reported in figure 1. It is possible that, while lymph nodal samples are more complex to assess, having more specific criteria compared to pulmonary lesions resulted in similar outcomes. We added the relevant speculations in the discussion section, as it follows:

DISCUSSION (lines 298-308): “Notably, since the adequacy criteria for lymph node samples are more precise compared to criteria for pulmonary lesions, we were expecting differences in terms of concordance according to the sampling site, as the pulmonologist had fewer reference points for defining an adequate pulmonary lesion, and the experience of a cyto-pathologist might have an edge; in fact, no significant difference was observed in terms of concordance between lymph nodal and non-lymph nodal lesions, suggesting a similar performance of the pulmonologist irrespective of collection site. One possible explanation can be found in the concept that non-lymph nodal lesions simply require less complex criteria since the adequacy of sample is defined by the presence of ab-normal cells, either neoplastic or non-neoplastic, while lymph nodes actually need more defined criteria due to their intrinsic complexity.”

REVIEWER 2

This is a very interesting and novel article about rapid On-Site Evaluation (ROSE) during bronchoscopy allows to assess sample adequacy for diagnosis and molecular analyses in the context of precision oncology.

Since this is a single center research, I encourage the authors to continue their work and perform a multi-center study to enable comparison of results obtained in different centers.

ANSWER: Thank you for your observations. Actually, having a multi-center study would provide more consistent and comprehensive information on the procedure, and eventually give insights on potentials and pitfalls of the training for ROSE across different hospitals. We included a specific sentence on this subject in the discussion section, as it follows:

DISCUSSION (lines 346-352): “In our opinion, the assessment of performance and future possibilities of ROSE deserves further studies, and to this regard a prospective, multicentric study assessing the performance of ROSE calls performed by pulmonologists of different institutions may help understanding the potential advantages and pitfalls of systematic practice of ROSE calls performed by trained pulmonologists, eventually with the additional aim of addressing the possible impact of different training modalities among the pulmonologists.”

COMMENTS BY THE EDITOR

To facilitate transparent and open scientific research, we encourage
authors to publish their results and experimental methodology in as much
detail as possible so that the results have a higher chance of being
reproduced. We have noticed that the main text of your manuscript is
quite brief, which may mean that the experiment, research background,
future research directions, or possible applications of the research are
not described in enough detail. For a detailed guide on Articles, please visit
https://www.mdpi.com/about/article_types

Please consider the following points in your revisions: add full
experimental details, present all the results completely, and describe a
comprehensive background to the research in the introduction section.In
addition, please provide the full name of the Institutional Review Board.

We would be grateful if you could send us your revised version before 10
July.

Thank you for your cooperation.

ANSWER: Thank you for your comments. Our article already includes the main results, but we generally increased the “methods” section, showing more details on the experimental design. This revision was made also in order to address the comments by Reviewer 1 and Reviewer 2. Additionally, we included more information on the current state of ROSE calls performed by non-pathologists in the Discussion section, improving the contextualization of our research. Finally, IRB details were included in the Methods section. We hope you will find the current form of the manuscript suitable for publication.

Reviewer 2 Report

Comments and Suggestions for Authors

This is a very interesting and novel article about rapid On-Site Evaluation (ROSE) during bronchoscopy allows to assess sample adequacy for diagnosis and molecular analyses in the context of precision oncology.

Since this is a single center research, I encourage the authors to continue their work and perform a multi-center study to enable comparison of results obtained in different centers.

Author Response

(The authors gave the same response as above.)
